# Salidroside, 8(*E*)-Nuezhenide, and Ligustroside from *Ligustrum japonicum* Fructus Inhibit Expressions of MMP-2 and -9 in HT 1080 Fibrosarcoma

**DOI:** 10.3390/ijms23052660

**Published:** 2022-02-28

**Authors:** Hojun Kim, Chang-Suk Kong, Youngwan Seo

**Affiliations:** 1Division of Convergence on Marine Science, Korea Maritime and Ocean University, Busan 49112, Korea; badguy47@naver.com; 2Department of Food and Nutrition, Silla University, Busan 46958, Korea; cskong@silla.ac.kr

**Keywords:** *Ligustrum japonicum* fructus, matrix metalloproteinases, MAPK, HT1080

## Abstract

A phenyl ethanoid, salidroside (SAL), and two secoiridoids, 8(*E*)-nuezhenide (NZD) and ligustroside (LIG), were isolated from fruits of *Ligustrum*
*japonicum,* used as traditional folk medicine, and their chemical structures were elucidated by the comparison of spectral data with published literature. Matrix metalloproteinases (MMPs) are major enzymes that play crucial roles in the metastasis and invasive behavior of tumors. In particular, MMP-2 and MMP-9, regulated by the MAPK signaling pathways, including p38, ERK and JNK, are known to play a key role in the degradation of the basement membrane. In the present study, the effects of SAL, NZD and LIG on the expression of MMP-2 and -9 were examined in phorbol 12-myristate 13-acetate (PMA)-induced HT 1080 cells. All the compounds significantly lowered the amount of MMP-2 and MMP-9 released, as determined by gelatin zymography and ELISA. In addition, the mRNA and protein expression levels of MMP-2 and MMP-9 were significantly suppressed, as measured by RT-PCR and Western blotting. According to the Western blotting assay, SAL and LIG effectively reduced the expression of MMP-2 in a dose-dependent manner. NZD lowered the expression of MMP-9 in a similar way. The phosphorylation of p38, ERK and JNK was also significantly suppressed by these compounds. These findings suggest that all the compounds regulate the release and expression of MMP-2 and MMP-9 via MAPK signaling pathways.

## 1. Introduction

Cancer is a fatal disease in humans, and many efforts have been made to treat it. Tumor progression is a complex multi-step process, involving cell division and growth, proteolysis of the extracellular matrix (ECM), cell migration and angiogenesis [1,2,3]. The extracellular matrix functions as a structural support for the cells, and plays a major role in cancer metastasis, tissue formation, intercellular communication and gene expression. Matrix metalloproteinases (MMPs) act on the degradation and remodeling of the extracellular matrix, and are also involved in the release of various signaling proteins [4,5]. Among the MMPs in the human body, MMP-2 and MMP-9, which are gelatinases, are known to digest type IV collagen, a major component of the extracellular matrix, and to induce cancer cell invasion and metastasis by participating in both basement membrane degradation and the formation of new blood vessels [6,7,8]. Therefore, because of the key role that MMPs play in tumorigenesis and metastasis, modulation of MMP expression could be a strategic target for the development of cancer-fighting and therapeutic methods.

Fibrosarcoma is a cancer that affects fibroblasts, which are responsible for making fibrous tissue found throughout the body. The highly tumorigenic HT1080 human fibrosarcoma cell line has been well studied and is widely preferred for studies on oxidative stress, inflammation and tumor cell invasion. In addition, HT1080 contains wild-type cancer-related proteins, which have been reported to be closely linked with inflammation and tumorigenic expressions [9].

Much attention has been given to MMPs over the past few decades. The pathogenesis of many major diseases, such as cardiovascular disease, cancer, inflammation, autoimmune diseases and neurodegenerative diseases, has been shown to be closely related to MMP function [10,11,12]. Despite considerable research efforts, the early clinical trials for first-generation MMP inhibitors have been disappointing, with undesirable side effects and a marked lack of efficacy. Therefore, targeting MMPs in the search for novel therapeutic agents against cancer is of great interest. Studies reported antitumor compounds that act on MMPs, including structure-based compounds, therapeutic antibodies and natural products. Natural products, which not only have structurally diverse chemistries, but have also been used for a long time as an important source of bioactive substances, have received great attention as a template for drug discovery, as well as a probe to investigate biochemical pathways [13,14,15].

In the course of screening inhibitors for MMP-2 and MMP-9 from natural resources, we found that an extract of *Ligustrum japonicum* fructus had an MMP inhibitory effect, and we recently reported that two compounds (GL-3 and oleonuezhenide) isolated from it significantly inhibited MMP-2 and MMP-9 [16,17]. In addition to GL-3 and oleonuezhenide, ^1^H NMR spectral analysis of the chromatographic fractions of the above plant extract showed that structurally related substances were present as minor components in the more polar fraction. Further separation on this polar fraction gave one phenylethanoid glycoside and two secoiridoid glycosides. In the current study, we report the isolation of salidroside (SAL), 8(*E*)-nuezhenide (NZD) and ligustroside (LIG), and their effects on the activation and expression of MMP-2 and MMP-9 in the HT1080 human fibrosarcoma cell line.

## 2. Results and Discussion

### 2.1. Structure Elucidation

The phenylethanoid, SAL, and the iridoids, NZD and LIG, were isolated from *Ligustrum japonicum* fructus (Figure 1). The chemical structures were elucidated and verified by 2D NMR spectroscopy, and by comparing the NMR spectral data with those previously published in the literature [18,19,20,21]. The 1H and 13C NMR spectral data of the isolated compounds were in accordance with the values found in the literature for the same compounds (Appendix A).

### 2.2. Cytotoxicity of the Isolated Compounds

Before the in vitro assays using the isolated compounds, their cytotoxicity in HT1080 cells was evaluated with the traditional 3-(4, 5-dimethylthiazol-2-yl)-2, 5-diphenyltetrazolium bromide (MTT) assay (Figure 2), with three different concentrations of 10, 50 and 100 µM. The survival rate of the HT1080 cells for all the compounds was over 80% at less than 100 µM, compared with the untreated control. These findings revealed that the tested samples were biocompatible with concentrations below 100 µM (Figure 2).

### 2.3. Effect of SAL, NZD and LIG on Enzymatic Activity of MMP-2 and MMP-9

The protein levels of active MMPs released by PMA-stimulated HT1080 cells, treated with or without SAL, NZD and LIG, were assessed based on enzymatic activity using the gelatin zymography assay. SAL and LIG significantly suppressed the levels of MMP-2 and MMP-9, as shown in Figure 3, but not in a dose-dependent manner. Furthermore, NZD showed the weakest inhibition effect against MMP-2 at 100 μM, but was even less effective in inhibiting MMP-9.

MMP-specific ELISA assays were performed to more clearly confirm the inhibitory effects of the compounds on the secretion of MMP-2 and MMP-9 into the cell culture media. In this case, unlike gelatin zymography, all the compounds showed strong inhibitory effects against MMP-2 and MMP-9, as shown in Figure 4; SAL, NZD and LIG, at a concentration of 100 µM, reduced the amount of MMP-9 to 383.3, 53.3 and 13.4 pq/mL, respectively, from 2888.6 pg/mL released by PMA stimulation only. Among the compounds, SAL exhibited the strongest inhibitory effect on MMP-9, which reduced the secretion of MMP-9 to 715.7 pg/mL, even at 10 µM. For MMP-2, all the compounds showed a greater reduction effect than for MMP-9, using the ELISA assay. SAL, NZD and LIG reduced the release of MMP-2 to 1.4, 2.6 and 2.0 ng/mL, respectively, lower than 4.6 ng/mL in the untreated blank cell, from 18.7 ng/mL after PMA stimulation at a concentration of 10 µM. The inhibition of MMP-2 and MMP-9 released into the cell culture was not concentration dependent. The secretions of MMP-9 at a 50 µM concentration of SAL, NZD and LIG were reduced to 11.4%, 18.3% and 13.9%, respectively, compared to the control. The secretions of MMP-2 were reduced to 17.5%, 14.0% and 10.7%, respectively, even at 10 µM. The IC_50_ of these compounds was expected to be comparable to doxycycline (Periostat), the (FDA)-approved MMP inhibitor for periodontal disease treatment, reported in the literature [12,22]. 

### 2.4. Effect of SAL, NZD and LIG on mRNA and Protein Expressions of MMP-2 and MMP-9 

In order to investigate the inhibitory mechanisms of all the compounds against the gelatinolytic activity of MMP-2 and MMP-9, the mRNA and total protein expression levels of MMP-2 and MMP-9 were evaluated by RT-PCR and immunoblotting (Figure 5). Contrary to the results obtained by gelatin zymography, when all the compounds were treated at a concentration of 100 µM, the mRNA expression of MMP-2 and MMP-9 was reduced to a level similar to that of the blank cell, not stimulated with PMA, indicating that they inhibit the expression of MMP-2 and -9 at the nuclear level (Figure 5a,b). A significant inhibitory effect of the compounds was also observed on the protein expression of MMP-2 and MMP-9. For MMP-9, SAL showed a protein expression level of 16.7%, lower than that of BK (23.4%), compared to the control, at a concentration of 100 μM, but for MMP-2, it exhibited expression levels of 35.5, 34.1 and 22.6%, similar to, or lower than, BK (35.3%) at 10, 50 and 100 μM concentrations, respectively (GAPDH was used as an internal standard for quantification). In contrast to SAL, NZD showed a lower protein expression pattern for MMP-9 than for MMP-2. For MMP-9, at concentrations of 10, 50 and 100 μM, NZD showed protein expression levels of 24.8, 15.1 and 11.5%, respectively, quite similar to that of BK (16.8%), and for MMP-2, a protein expression level of 50.5% (BK 35.3%) at 100 μM was shown. In the case of treatment with LIG, MMP-9 protein expression was only significantly suppressed at a concentration of 100 μM, and a 49.1% expression level (BK 23.4%) was observed. MMP-2 protein expression was effectively suppressed at all the treated concentrations of LIG, in a dose-dependent manner, showing expression levels of 57.2, 32.9, 20.2 and 15.3% at concentrations of 1, 10, 50 and 100 μM, respectively (Figure 5c,d).

### 2.5. Effect of SAL, NZD and LIG on MAPK Signaling Pathway

It is known that activation of the mitogen-activated protein kinase (MAPK) pathway is part of the regulation of tumor metastasis, and suppression of its activation is closely related to the downregulation of MMP expression [23,24,25,26]. The phosphorylation of MAPK-associated proteins, i.e., p38, p-ERK and p-JNK, was evaluated to elucidate the mechanisms of the intracellular pathways involved in MMP inhibition. As expected, PMA stimulation resulted in increased expression of phosphorylated ERK, JNK and p-38, in association with increased cellular activity. Activation of the MAPK pathway, stimulated by PMA, is shown in Figure 6. The phosphorylation level of the MAPK pathway was measured by the amount of protein expressed in the Western blotting assay. All the compounds significantly attenuated the protein expressions of p-p38, p-ERK and p-JNK, when compared to the control, at concentrations of more than 50 μM. In particular, SAL and LIG showed 47.2% and 57.5% expression levels of p-JNK, respectively, even at a concentration of 1 μM, and NZD showed an expression level of 28.7% of p38 at the same concentration (Figure 6a,b). Hence, it could be suggested that the expression levels of MMP-2 and -9 were suppressed by the downregulation of p-ERK, p-JNK and p-p38 MAPK-mediated pathways. The results matched well with the mRNA and protein expressions of MMP-2 and MMP-9.

MMP-2 and MMP-9 are proteolytic enzymes that degrade the extracellular matrix; they are closely related to the metastasis and invasion of tumors, and have long been regarded as attractive targets for the development of cancer therapeutics [27,28,29]. However, since most synthetic MMP inhibitors are toxic, the need for the development of more selective and newer pharmacophores of MMP inhibitors, without side effects, has been raised [12,14]. Natural plant extracts and their phytochemicals are recognized as promising and rich sources for achieving chemoprevention that reduces the morbidity and mortality of cancer patients, as well as for discovering novel lead structures with inhibitory activity against MMPs [13,15].

SAL, previously isolated from several plants, is a member of the phenylethanoid glycoside class. It has been reported to show a variety of bioactivities, such as antibacterial, anticancer, anti-inflammatory and anti-fertility. In this study, a remarkable decrease in MMP-2 and -9 activity by SAL was observed in HT1080 cells. The inhibitory effect of MMP-2 and -9 activity by SAL was previously measured in human lung cancer cells (A549) and human bladder cancer cells (T24), respectively, and a significant decrease in activity has been reported [30,31]. In a study by Sun et al. [32], SAL was also shown to increase the tissue inhibitor of metalloproteinase-2 (TIMP-2), which is a crucial regulator of MMP-2 and MMP-9 activation. It was reported that salidroside treatment increased TIMP-2 levels in conditioned medium of HT1080 cells. The effect of SAL on the TIMP-2 levels might be credited for the differences between gelatin zymography and ELISA assays, where the former detects both pro and active enzymes. Although MAPK suppression was suggested as the mechanism for suppressing MMP expression, its effect on the TIMP-2 levels hints at a concurrent mechanism to hinder MMP activity.

Iridoid glycosides are monoterpenoid glycosides with a characteristic cyclopentanopyran part. The cleavage of the bond between C-7 and C-8 in the cyclopentane ring of iridoids leads to secoiridoids having carboxylic acid and olefin moieties. A number of biological activities have been published for secoiridoid glycosides, including antioxidant, anti-diabetic, anti-inflammatory, immunomodulatory, neuroprotective and anti-cancer effects [33]. As already described, NZD and LIG showed significant inhibitory effects on the enzymatic activity of MMP-2 and -9. Two previous studies reported that secoiridoid glycosides suppressed MMP enzymes [34,35], but, with the exception of GL-3 and oleonuezhenide reported by our group, there have been no reports on the inhibitory activity of secoiridoid glycosides against MMP-2 and -9 in actual experiments [15]. According to the literature, the MMP-9 inhibitory effect of secoiridoid glycosides was calculated by molecular docking, through computer simulations. The computational molecular docking analysis revealed that LIG was effective in preventing MMP-1, MMP-3 and MMP-9 activities [35]. The possible inhibition of the direct enzymatic activity of MMPs further suggests a concurrent mechanism for LIG and NZD, aside from MAPK suppression, similar to SAL and its effect on TIMPs. This might explain the differences that were observed in the gelatin zymography and expression assays, in terms of effectiveness, between the compounds. In this context, future studies focusing on the effects of these compounds on TIMP levels and direct enzymatic activity would provide invaluable inputs regarding the potential of SAL, NZD and LIG as antitumor agents.

## 3. Materials and Methods

### 3.1. Apparatus and Reagents

The ^1^H and^13^C NMR spectra were recorded on a Varian NMR 300 spectrometer (300 MHz for ^1^H and 75.5 MHz for ^13^C). Chemical shifts (δ in ppm) were referenced to the residual solvent peak. The NMR solvent used was CD_3_OD (Cambridge Isotope Laboratories, Inc., Cambridge, MA, USA, deuterium degree 99.95%). Mass spectroscopic data were obtained using high-resolution ESI mass spectrometer at the Korean Basic Science Institute (Center of Research Equipment, Ochang, Chungbuk, Korea). High-performance liquid chromatography (HPLC) was performed with a Dionex P580 equipped with Varian 350 RI detector using a column (YMC pack ODS-A, 250 × 10 mm, S 5 μM, 12 mm) and a guard column (7.5 × 4.6 mm, Alltech). All solvents used were of spectral grade or were distilled from the glass before use.

### 3.2. Extraction, Fractionation and Isolation

The air-dried samples of *Ligustrum japonicum* fructus were purchased from an online market (Omniherb Company, Daegu, Korea) in 2014. The fruits were extracted twice for 24 h using dichloromethane, and then further extracted twice using methanol (MeOH). The crude extracts obtained through dichloromethane and MeOH were combined (197.6 g), and the combined extracts were partitioned between methylene chloride and water (H_2_O). The methylene chloride layer was further partitioned between *n*-hexane and 85% aq.MeOH, and the water layer was also further partitioned between *n*-butanol (*n*-BuOH) and H_2_O. These four solvent fractions were concentrated on a rotary evaporator to give *n*-hexane (56.1 g), 85% aq.MeOH (46.3 g), *n*-BuOH (33.0 g) and H_2_O (51.5 g), respectively. A portion of the *n*-BuOH (10.0 g) fraction was chromatographed onto a C_18_ silica gel with MeOH-H_2_O solvent gradient in decreasing polarity (50, 60, 70, 80, 90% aq.MeOH, and 100% MeOH). Fraction 2 (60% aq.MeOH, 2.93 g) was further chromatographed by HP20 column chromatography to give 5 fractions (100% H_2_O, 50% aq.MeOH, 50% aq.acetone, 100% MeOH and 100% acetone), named as HP1 to HP5, respectively. The HP2 (50% aq.MeOH, 0.532 g) fraction was separated by reversed phase HPLC (ODS-A, 20% aq.MeOH) to afford SAL (7.8 mg), NZD (7.9 mg) and LIG (5.6 mg).

Salidroside (SAL): ^1^H NMR (300 MHz, CD_3_OD) δ 6.95 (2H, H-2/-6), 6.68 (2H, H-3/-5), 5.03 (1H, H-1′), 3.70~3.79 (5H, H-8/-2′/-5′/-6′), 3.54 (1H, H-6′), 3.49 (1H, H-3′), 3.40 (1H, H-4′), 2.72 (2H, H-7); ^13^C NMR (75 MHz, CD_3_OD) δ 156.6 (C-4), 130.8 (C-2/-6), 130.5 (C-1), 116.0 (C-3/-5), 104.3 (C-1′), 78.0 (C-5′), 77.9 (C-3′), 75.0 (C-2′), 72.1 (C-8), 71.6 (C-4′), 62.7 (C-6′), 36.4 (C-7); ESI-MS (positive-ion mode) *m*/*z*: 301 [M+H]^+^.

8(*E*)-nuezhenide (NZD): ^1^H NMR (300 MHz, CD_3_OD) δ 7.15 (1H, H-3), 6.95 (2H, H-2″/-6″), 6.68 (2H, H-3″/-5″), 5.36 (1H, H-8), 5.03 (2H, H-1′/-1‴) 4.34 (1H, H-4‴), 4.27 (1H, H-5‴), 4.09 (1H, H-4‴), 3.73~3.79 (3H, H-3′/-5′/-2‴), 3.76 (3H, H-11-COOCH_3_), 3.70 (1H, H-α), 3.54 (2H, H-6′), 3.49 (3H, H-5/-3′/-3‴), 3.40 (2H, H-4′/-4‴), 2.72 (2H, H-β), 2.41 (2H, H-6), 1.71 (3H, H-10); ^13^C NMR (75 MHz, CD_3_OD) δ 172.8 (C-7), 168.5 (C-11), 156.6 (C-4″), 155.0 (C-3), 130.8 (C-2″/-6″), 130.5 (C-1″), 130.3 (C-9), 124.8 (C-8), 116.0 (C-3″/-5″), 104.3 (C-1‴), 100.7 (C-1′), 95.0 (C-1), 78.4 (C-5′), 77.9 (C-3‴), 77.8 (C-3′), 75.1 (C-5‴), 74.9 (C-2‴), 74.7 (C-2′), 72.2 (C-α), 71.5 (C-4‴), 71.4 (C-4′), 65.0 (C-6‴), 62.7 (C-6′), 52.0 (C-11-COOCH_3_), 41.3 (C-6), 36.4 (C-β), 31.8 (C-5), 13.8 (C-10); ESI-MS (positive-ion mode) m/z: 687 [M+H]^+^.

Ligustroside (LIG): ^1^H NMR (300 MHz, CD_3_OD) δ 7.15 (1H, H-3), 6.95 (2H, H-2″/-6″), 6.68 (2H, H-3″/-5″), 5.53 (1H, H-1) 5.36 (1H, H-8), 5.03 (1H, H-1′), 4.41 (2H, H-α), 3.76 (3H, H-11-COOCH_3_), 3.40~3.79 (6H, H-5/-2′/-3′/-4′/-5′/-6′), 2.83 (2H, H-β), 2.41 (2H, H-6), 1.72 (3H, H-10); ^13^C NMR (75 MHz, CD_3_OD) δ 173.0 (C-7), 168.4 (C-11), 156.9 (C-4″), 154.9 (C-3), 130.9 (C-2″/-6″), 130.0 (C-9), 129.9 (C-1″), 124.7 (C-8), 116.1 (C-3″/-5″), 100.7 (C-1′), 95.0 (C-1), 78.4 (C-5′), 77.9 (C-3′), 74.7 (C-2′), 71.4 (C-4′), 66.9 (C-α), 62.7 (C-6′), 51.9 (C-11-COOCH_3_), 41.2 (C-6), 35.2 (C-β), 31.9 (C-5), 13.6 (C-10); ESI-MS (positive-ion mode) *m*/*z*: 525 [M+H]^+^.

### 3.3. Cell Maintenance and Determination of Cytotoxicity

Human fibrosarcoma cell line, HT1080, was adopted from KCLB (Korea Cell Line Bank, Jongro, Seoul, Korea), and cultured in T-75 flasks (SPL, Pocheon, Gyeonggi, Korea) in 5% CO_2_ and at 37 °C in a humidified incubating condition, and the medium used was RPMI 1640 (GenDEPOT, Baker, TX, USA) with 10% fetal bovine serum (FBS) (GenDEPOT, Baker, TX, USA) and 100 unit/mL penicillin–streptomycin (Gibco-BRL, Grand Island, NY, USA). The cell lines were washed with PBS buffer (Gibco-BRL, Grand Island, NY, USA) and the medium was changed six times a week.

For cell viability assay, cell lines were transferred to 96-well plates at 5 × 10^3^ cells/well density. After their transfer, cells were cultured for 24 h and their medium was changed with fresh medium, which was followed by the treatment with samples (100, 50 and 10 µM). Cells were re-supplied with fresh medium after 24 h of incubation and treated with 100 µL of MTT solution (1 mg/mL), and further incubated for 4 h. Finally, the wells were aspirated and introduced into 100 µL of dimethyl sulfoxide (DMSO), in order to solubilize the formazan crystals for the measurement by a Victor3 reader (PerkinElmer, Waltham, MA, USA) at 540 nm optical density.

### 3.4. Determination of Active MMP-2 and MMP-9 Levels by Gelatin Zymography

The protein levels of active MMP-9 and MMP-2 secreted from the HT1080 cells were evaluated by gelatin zymography. Cells were transferred to 24-well plates at 2 × 10^5^ cells/well density. The cells were cultured for 24 h, and, following this, the medium was replaced with serum-free medium. Next, the cells were treated with samples at different concentrations. After 1 h, phorbol 12-myristate 13-acetate (PMA, 10 ng/mL) was added to each well, in order to stimulate MMP expression, and cells were further incubated for 24 h. The culture supernatant was harvested, and total protein content of the culture supernatant was evaluated by Bradford protein assay. Conditioned supernatants with the same amount of protein were loaded onto 10% sodium dodecyl sulfate-polyacrylamide gel containing 1.5 mg/mL gelatin (Sigma Aldrich, St. Louis, MO, USA). Electrophoresis was conducted under non-reducing conditions. Polyacrylamide gels were washed by buffer (2.5% Triton X-100 (JUNSEI, Japan), 50 mM Tris-HCl, pH 7.5) for 30 min to remove sodium dodecyl sulfate. Then, gels were kept in developing buffer (10 mM CaCl_2_, 50 mM Tris-HCl, 150 mM NaCl) to promote the digestive activity of MMPs on gelatin. Clear zones of gelatin hydrolyzation by MMPs were observed against the Coomassie blue-stained background obtained via incubation in staining buffer (Biosesang, Seongnam, Gyeonggi, Korea). The clear zones were imaged with CAS-400SM Davinch-Chemiimager^TM^ (Davinch-K, Geumcheon, Seoul, Korea).

### 3.5. Enzyme-Linked Immunosorbent Assay (ELISA) for the Detection of MMP-2 and MMP-9 Release

For the detection of MMP-2 and MMP-9 levels in the HT1080 culture medium, the cells were cultured in 6-well plates and incubated for 24 h, washed with PBS, and then stimulated with PMA. Thereafter, cells were incubated for 24 h in the presence or absence of the sample. MMP-2 and MMP-9 proteins secreted into the culture supernatant were detected with an ELISA kit (R&D systems, Inc., Minneapolis, MN, USA) according to the manufacturer’s instructions.

### 3.6. RNA Extraction and Reverse Transcription Polymerase Chain Reaction (RT-PCR) Analysis

Whole RNA from HT1080 cell lines was isolated by Trizol reagent (Invitrogen Co., Waltham, MA, USA). Isolated RNA (2 μg) was added to RNase-free water and oligo dT, denatured at 70 °C for 5 min and cooled at 4 °C immediately. In addition, RNA was reverse transcribed in mixtures containing 1X RT buffer, 1 mM dNTP, 500 ng oligo dT, 140 U M-MLV reverse transcriptase and 40 U RNase inhibitor at 42 °C for 1 h and at 72 °C for 5 min, using thermal cycler (Bio-rad, USA). Each target DNA was amplified by the following sense and antisense primers: forward 5′-TGA-AGG-TCG-GTG-TGA-ACG-GA-3′ and reverse 5′-CAT-GTA-GCC-ATG-AGG-TCC-ACC-AC-3′ for MMP-2; forward 5′-CAC-TGT-CCA-CCC-CTC-AGAGC-3′ and reverse 5′-CAC-TTG-TCG-GCG-ATA-AGG-3′ for MMP-9; forward 5′-GCC-ACC-CAG-AAG-ACT-GTG-GAT-3′ and reverse 5′-TGG-TCC-AGG-GTT-TCT-TAC-TCC-3′ for β-actin. Target DNA amplification cycles were performed at 95 °C for 45 s, 60 °C for 1 min and 72 °C for 45 s. After 30 cycles, the DNA products were separated by electrophoresis on 1.5% agarose gel at 100 V for 30 min. Finally, gels were stained with 1 mg/mL EtBr solution and visualized under UV light using CAS-400SM Davinch-Chemiimager^TM^ (Davinch-K, Geumcheon, Seoul, Korea).

### 3.7. Western Blot Analysis

Protein levels of MMPs were quantified by Western blot assay. Whole proteins from HT1080 cell lines were isolated by RIPA buffer (Sigma Aldrich, St. Louis, MO, USA). Lysed cell lines were loaded onto 12% sodium dodecyl sulfate-polyacrylamide gel and electrophoresis was conducted at 100 V for 90 min. Then, separated proteins were transferred onto a nitrocellulose transfer membrane (Whatman^®^, Maidstone, Kent, UK), blocked with 5% skim milk (BD, Franklin Lakes, NJ, USA) in 1X TBST buffer (LPS solution, Daedeok, Daejeon, Korea), incubated with primary antibodies (MMP-9, MMP-2 and β-actin) (Cell signaling, Danvers, MA, USA) overnight, and then re-incubated with subsequent secondary antibodies at room temperature for 1 h. Immunoreactive proteins were detected using Western Blotting Detection Reagents kit (GE Healthcare, Chicago, IL, USA) and protein bands were observed using a CAS-400SM Davinch-Chemi imager^TM^ (Davinch-K, Geumcheon, Seoul, Korea).

### 3.8. Statistical Analysis

All numerical values that were presented in the current study were the mean of three independent experiments ± SD. The statistically significant differences between the different groups were determined by employing one-way analysis of variance (ANOVA) using Statistical Analysis System software v9.1 (SAS Institute, Cary, NC, USA). ANOVA was coupled with Duncan’s multiple range test as post hoc analysis. The difference between the different groups was accepted as statistically significant when *p* < 0.05.

## 4. Conclusions

In the current study, the MMP inhibitory effects of SAL, NZD and LIG were estimated in PMA-stimulated HT1080 cells. The results of this study showed that these compounds are potentially effective blockers of tumor cell migration and metastasis, by inhibiting MMP-2 and MMP-9 expression and activation through regulation of the MAPK signaling pathway. Therefore, the findings and reports from our two recent studies support the fact that these compounds may be potential candidates for the prevention and treatment of tumor progression [34,35]. In the future, further studies are needed to evaluate the MMP inhibitory effects of various derivatives of secoiridoid glycoside, and to understand their mechanisms of action.

## Figures and Tables

**Figure 1 ijms-23-02660-f001:**
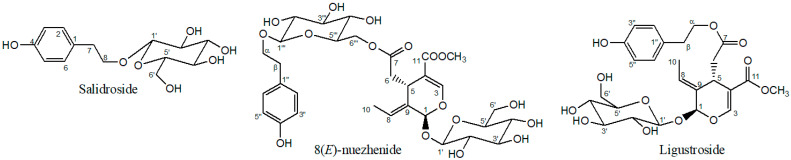
Chemical structures of isolated compounds salidroside (SAL), 8(*E*)-nuezhenide (NZD) and ligustroside (LIG).

**Figure 2 ijms-23-02660-f002:**
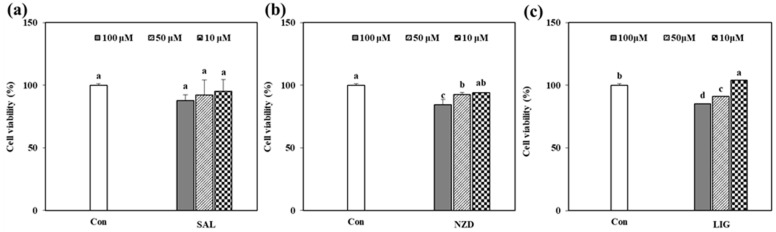
Effect of SAL (**a**), NZD (**b**) and LIG (**c**) on viability of HT1080 cells. The data are presented as mean ± standard deviation of three independent experiments for each compound (Con: untreated cells). ^a–d^ Means with different letters indicate statistically significant difference at *p* < 0.05 level, as assessed by Duncan’s multiple range test.

**Figure 3 ijms-23-02660-f003:**
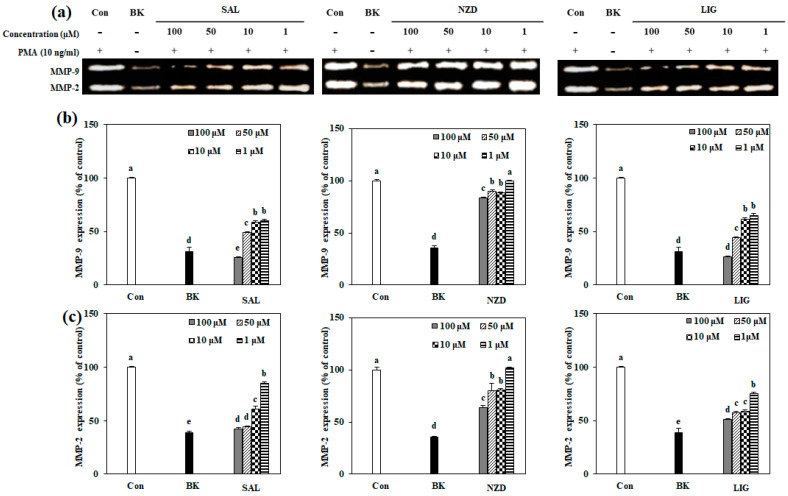
Effects of SAL, NZD and LIG on the levels of MMP-9 and MMP-2, evaluated by gelatin zymography in HT1080 cells (Con: untreated PMA-stimulated cells; BK: untreated unstimulated cells). (**a**) Gelatinolytic activities of MMP-9 and MMP-2 on gelatin containing 10% polyacrylamide gel. White areas in gelatin zymography, which represent digestion by active MMP-9 (**b**) and MMP-2 (**c**), were quantified by densitometry. Control and blank were without treated samples and blank was also without PMA. ^a–e^ Means with different letters at the same concentration are significantly different (*p* < 0.05), as assessed by Duncan’s multiple range test.

**Figure 4 ijms-23-02660-f004:**
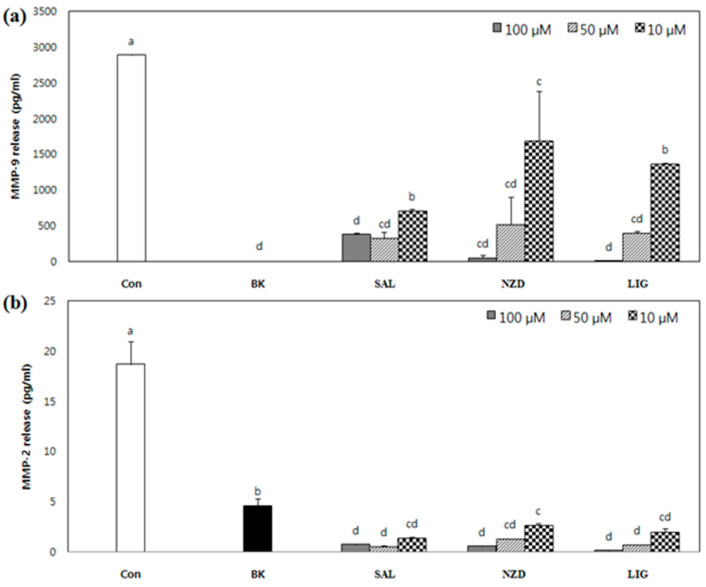
Released amounts of MMP-9 (**a**) and MMP-2 (**b**) by PMA-stimulated HT1080 cells treated with or without SAL, NZD and LIG at given concentrations (Con: untreated PMA-stimulated cells; BK: untreated unstimulated cells). The culture medium was collected and analyzed for the MMP-2 and -9 levels by ELISA. Data are mean ± SD of three replicates (*n* = 3). ^a–e^ Means with different letters at the same concentration are significantly different (*p* < 0.05), as assessed by Duncan’s multiple range test.

**Figure 5 ijms-23-02660-f005:**
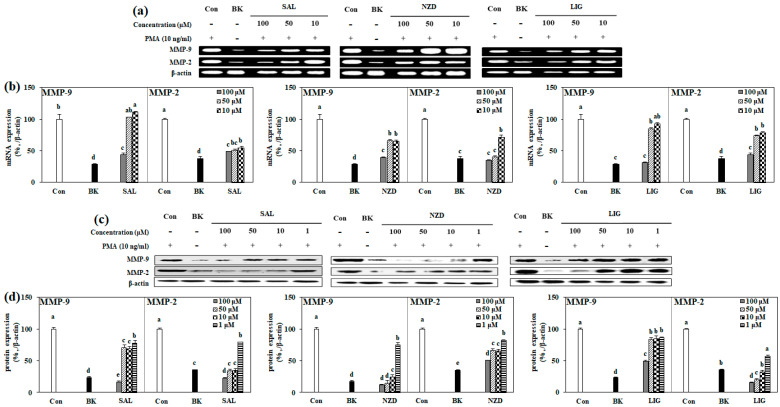
Effects of SAL, NZD and LIG on mRNA (**a**,**b**) and protein (**c**,**d**) expression levels of MMP-9 and MMP-2, analyzed by RT-PCR and immunoblotting in HT1080 cells, respectively (Con: untreated PMA-stimulated cells; BK: untreated unstimulated cells). β-actin was used as the loading control. HT1080 cells were treated with compounds prior to stimulation with PMA. The mRNA and protein expression levels of MMP-9 and MMP-2 were densiometrically calculated and given as a relative percentage of the stimulated untreated control group (Con). Expression levels were normalized against housekeeping control, β-actin. ^a–e^ Means with different letters at the same concentration are significantly different (*p* < 0.05), as assessed by Duncan’s multiple range test.

**Figure 6 ijms-23-02660-f006:**
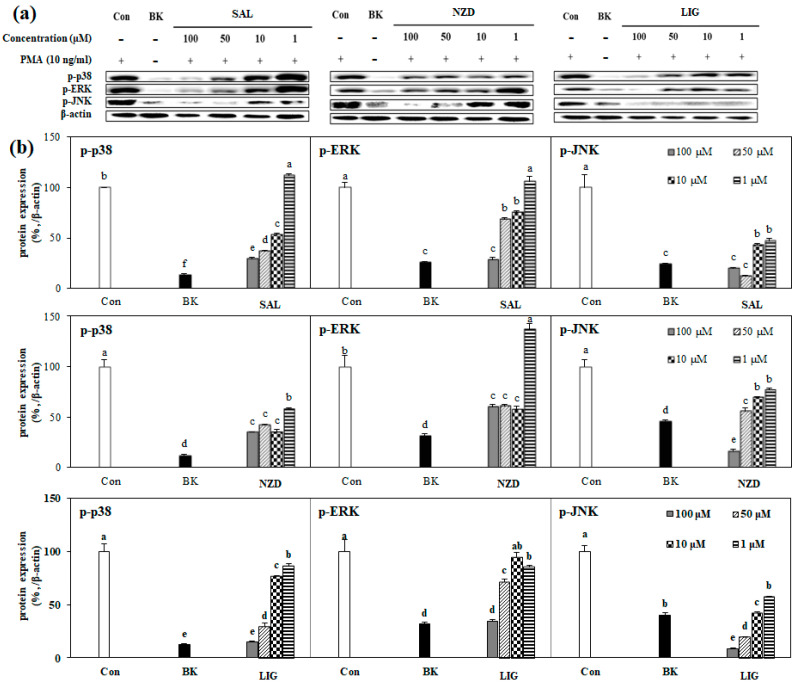
(**a**) Effects of SAL, NZD and LIG on protein levels of phosphorylated (p-) p38, ERK and JNK by Western blot in HT1080 cells (Con: untreated PMA-stimulated cells; BK: untreated unstimulated cells). β-actin was used as the loading control. The MMP production in cells was stimulated with phorbol 12-myristate 13-acetate (PMA), and cells were subjected to treatment with isolated compounds. The phosphorylated protein levels were densiometrically (**b**) calculated from band images and given as relative percentage of PMA-stimulated untreated control group (Con). Expression levels were normalized against house-keeping control, β-actin. ^a–e^ Means with different letters at the same concentration are significantly different (*p* < 0.05), as assessed by Duncan’s multiple range test.

## Data Availability

The data presented in this study are available on request from the corresponding author.

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
