# Peer review of "Salidroside, 8(E)-Nuezhenide, and Ligustroside from Ligustrum japonicum Fructus Inhibit Expressions of MMP-2 and -9 in HT 1080 Fibrosarcoma"

_ijms, 2022, doi:10.3390/ijms23052660_

Round 1

Reviewer 1 Report

The article by Kim, H. et al. about MMPs expression inhibition in fibrosarcoma cells appears scientifically correct. Manuscript formatting, with results and discussion after the introduction seems a bit strange for me, and some clues mentioned in minor issues suggest that the text was originally conceived in the traditional format (introduction-materials-results-discussion-conclusion), but it´s just an opinion. Authors performed a nice experiment, meritorious for publication in my opinion.

The manuscript has a generally fine grammar / style. It would probably benefit from a review to correct some expressions (some are mentioned as minor issues), but I don´t want to be critical on this regard. By the way, there is a character I cannot identify, located after some numbers referring to concentration, please check.

There are major issues:

  • In the abstract and in the whole text, the studied compounds, salidroside, 8(E)-nuezhenide, and ligustroside, are referred as (1), (2) and (3) respectively. I find it really confusing, as long as one would think they are references. Please, employ acronyms or abbreviations, for example SAL, NZD and LIG. After defining these acronyms the first time, you should use this form in the whole text for consistency reasons.
  • I would like to see why did you choose a fibrosarcoma cell and that particular line. A new paragraph about this in the introduction section would be fine to enhance the manuscript. You can also move some information from the “discussion” section to improve the introduction (see minor issues below).

There are some minor issues as well:

  • Introduction, page 1, lines 27-29. Growth and metastasis of cancer is a complex phenomenon. It is not as simple as presented in the first sentence of the work. You should better calibrate this sentence.
  • Introduction, page 1, lines 30-32. The matrix functions AS structural support, playing major roles in cancer metastasis, etc.
  • Results, page 2, line 73 and Materials, page 8, line 260. Here MMT is mentioned for the first time in results, but it is defined in the materials section. If you want to define this acronym, you must do it the first time it is mentioned (this point would be correct if you use the second section of the text for materials and the third for results). You may also avoid defining it if you prefer.
  • Results, Figure 2. Here is a good example of the confusion generated by the use of mere numbers as abbreviation for the different compounds. Please, correct it and use a new abbreviation or acronym. However, there are more flaws in this figure. You state in the legend that data are presented as mean ± SD, but in the graph data are presented as mean +SD, the lower part of SD is omitted. In addition it is written “group” in the legend, but I think it would be more precise to employ “compound”.
  • Results, page 3, Figure 3. I suppose “Con” is control and “BK” is blank, but you should specify in the legend. They were without treatMENT.
  • Results, page 5, Figure 5. The weft filling the bars is in some cases difficult to distinguish.
  • The previous assessment can be applied to subsequent figures and text, which I find not enough clearly presented. I think a revision here is needed.
  • Discussion, page 5, lines 168-170. You begin the introduction and the discussion with the same assessment. The first paragraph of the discussion could also fit in the introduction section.
  • Actually the whole discussion (I think lines 168-199 are similar to a traditional discussion section) could perfectly fit in the introduction. You make a very good description, but you don´t discuss your results, connecting them with the existing literature.
  • Materials, page 7, line 212. Were these samples purchased on-line or in a market? Might these samples were purchased online in a South Korean store or market? You should be more clear / specific.
  • Materials, page 7, lines 227-248. Could you make this text easier to read? You could also consider to include Figure 1 here, but as said previously, I think all the manuscript would be more logical if you include materials and methods section before results. May be I´m old fashioned in this regard…
  • Materials, page 8, line 252. You repeat “medium” twice, please remove one.
  • Materials, page 8, lines 257-258. “After their transfer, cells were cultured 24h and their medium was changed with fresh…
  • Materials, page 8, line 261. Solution was removed. Solutions were removed.
  • Materials, page 8, line 263. A Victor3 reader.
  • Materials, page 8, lines 269-271. PMA was added to enhance.
  • Materials, page 8, lines 274-275. Electrophoresis was then conducted under non-reducing conditions.
  • Materials, page 8, lines 283-284. Cells were pre-incubated, washed and then, stimulated.
  • Materials, page 9, lines 291-292. You shouldn´t begin a sentence with “And”. In addition, you don´t need to include an “a” before a plural form of a substantive (mixtures in this case).
  • Materials, page 9, lines 305-314. This paragraph has the same style issues mentioned previously.

Author Response

The reviewers’ comments were highly valuable, and the manuscript was modified accordingly. Changes made were highlighted in red color in the manuscript. All the co-authors have read and accepted the revised version of the manuscript. Your consideration of this manuscript for publication in “International Journal of Molecular Sciences” would be greatly appreciated. We would like to appear for further communication in this regard. 

Reviewer 2 Report

I have the following issues that should be addressed before this manuscript is accepted for publication:

(1) One major concern with this paper is the physiological relevancy of these compounds studied. Sure you have shown that these compounds reduce the expression and activity of specific MMPs, but this does not necessarily translate to reduced invasive behavior. Some simple 3D invasion assays should be performed to test if reduction in these MMPs translates to reduces invasive capacity. It has been shown in other studies that cells can compensate for reduced specific MMP expression through the up-regulation of other MMPs or ADAMs. Without this data, the present study is rather limited and superficial in depth. 

(2) In Figure 5, I am not sure what the difference is between the Western blots in panel A and panel C aside from the lack of a 1uM concentration in panel C. Furthermore, why are the Western blot images presented differently between these two panels? One is a negative image and one is not. The blots should all be shown the same across all figures.

(3) In all the figures, it might be helpful for the reader to include the names of the compounds instead of just writing 1, 2, 3, etc. 

(4) The nomenclature of "blank" used in the figures doesn't seem appropriate. Perhaps something like "unstimulated" would be more appropriate. 

(5) Throughout the manuscript, there are weird spiral icons that appear in the text. They appear to be errors where the Greek letter micro was supposed to be. 

(6) A minor English grammar editing is necessary to improve the ease of read of this manuscript.

Author Response

(The authors gave the same response as above.)

Round 2

Reviewer 2 Report

The authors have addressed all of my concerns that I had originally had.

Author Response

We sincerely thank you for your valuable comments. Manuscript was revised accordingly, and the changes were highlighted with red font for your convenience. In detail, the instances where zymography results were mistakenly interpreted as an effect on enzymatic activity were replaced with indication of protein levels. Also, per your suggestion, the Results and Discussion section was improved with discussions (Line 196-203 and 217-222) focusing on the effect of compounds on TIMPs and direct enzymatic activity referring published literature since conducting new experiments was not possible at this point. The abbreviations were also implemented where it was missed in previous revision. Also, minor mistakes were eliminated to the best of our knowledge. Again, we thank you for your efforts and humbly hope that the manuscript is suitable for publication at its current state.